# Cytokine Expression by Human Macrophage-Like Cells Derived from the Monocytic Cell Line THP-1 Differs between Treatment with Milk from Preterm- and Term-Delivering Mothers and Pasteurized Donor Milk

**DOI:** 10.3390/molecules25102376

**Published:** 2020-05-20

**Authors:** Veronique Demers-Mathieu, Robert K. Huston, David C. Dallas

**Affiliations:** 1Nutrition Program, School of Biological and Population Health Sciences, College of Public Health and Human Sciences, Oregon State University, Corvallis, OR 97331, USA; 2Department of Pediatrics, Randall Children’s Hospital at Legacy Emanuel, Portland, OR 97227, USA; Robert_Huston@mednax.com (R.K.H.); Dave.Dallas@oregonstate.edu (D.C.D.)

**Keywords:** tumor necrosis factor-α, interleukin-6, interleukin-10, preterm infant, immunomodulatory effect, immature immunity, immune response, newborn

## Abstract

Immunomodulatory proteins from human milk may enhance the protection and development of the infant’s gut. This study compared the immunomodulatory effects of treatment with milk from preterm-(PM) and term-delivering (TM) mothers and pasteurized donor milk (DM) on cytokine gene expression in human macrophage-like cells derived from the monocytic cell line THP-1. The gene expression of tumor necrosis factor-α (TNF-α), interleukin (IL)-6, IL-12 (p40), IL-10 and GAPDH in macrophages treated with PM, TM and DM at steady and activated (inflammatory) states were measured using real-time reverse transcription-polymerase chain reaction. TNF-α and IL-6 in macrophages (both states) with DM were higher than PM or TM. IL-10 in steady state macrophages with DM was higher than PM whereas DM increased IL-10 in activated macrophages compared with TM. TM increased IL-6 and IL-12 (p40) in steady state macrophages compared with PM. IL-12 (p40) in activated macrophages with TM was higher than PM. IL-10 in steady state macrophages with TM was higher than PM. These results suggest that DM induces higher gene expression of pro-inflammatory and anti-inflammatory cytokines in macrophages compared with PM or TM. PM reduced gene expression of pro-inflammatory cytokines compared with TM, which may decrease the development of necrotizing enterocolitis and systematic inflammation.

## 1. Introduction

Mother’s milk contains numerous immune proteins (e.g., antibodies, lactoferrin and cytokines) and immune cells (e.g., macrophages, neutrophils, plasma cells, B cells and T cells) that enhance the protection and development of the infant immune system [1,2]. Preterm infants fed exclusively breast milk have lower risk of necrotizing enterocolitis (NEC) [3], infections [4] and sepsis [5] compared with those fed bovine-based milk products. This reduced risk is partially related to their exposure to the immune proteins from human milk. Another explanation of the elevated risk of infection in the first 2 months of postnatal age is that preterm infants receive lower levels of maternal transplacental pathogen-specific IgG than term infants [6]. Therefore, human milk immune components are critical for supporting the infant while its own innate and adaptive immune system is immature.

Microbial infections are one of the major causes of mortality in newborn infants due to the deficiency in antibody production and cell-mediated immunity, including the mononuclear phagocytes [7]. Mononuclear phagocytes are blood monocytes and tissue macrophages that produce cytokines to regulate immune responses. Pro-inflammatory cytokines tumor necrosis factor-α (TNF-α), interleukin (IL)-6 and IL-12 are produced by mononuclear phagocytes and both induce inflammatory responses that may help to fight against infectious agents [8]. TNF-α can induce death in human fibroblast cell lines and activate neutrophils. IL-6 can induce the proliferation and antibody secretion of B-cells. IL-12 can modulate the adaptive immune system by inducing the differentiation of helper T cells to the T_H_1 subset [8]. Some type of mononuclear phagocytes, including macrophages, produce the anti-inflammatory cytokine interleukin 10 (IL-10), which regulate epithelial cell integrity in the small intestine [9]. IL-10 can enhance the proliferation of B cells and their secretion of IgA and inhibits T_H_1 cell generation [10]. In addition to these functions, these cytokines have other cell modulating activities (pleiotropic effects).

Human milk contains TNF-α, IL-6 and IL-10 that are secreted by macrophages and by mammary epithelial cells [11,12]. Higher concentrations of these cytokines in human milk are correlated with higher macrophages counts [12,13]. Cytokines IL-1β, interferon (IFN)-γ and macrophage colony-stimulating factor (M-CSF) were also detected in human milk [7,14]. Cytokines present in human milk might compensate for the developmental delay of the neonatal immune system.

Newborns possess less capacity to produce cytokines, which could affect the activation of their immune responses. Cytokine production or their gene expression by neonatal blood mononuclear cells and T cells was slightly (TNF-α) or strongly (granulocyte-M-CSF, IL-4, IL-10 and IFN-γ) lower compared with adult cells [15,16,17]. This lower cytokine secretion by phagocyte cells resulted in a lower antibody response during microbial infection in newborns [18]. Human milk cytokines could help simulate infant intestinal immune cells, which could prevent infection.

Previous clinical studies have compared the cytokine production in blood between preterm and term infants. Monocytes isolated from infant blood and then tested with stimulation of lipopolysaccharide (LPS) in vitro. The whole blood IL-6 production induced by bacterial pathogens (group *B streptococci*, *Escherichia coli* or *Streptococcus pneumonia*) in preterm infants (born before 30 weeks of gestation) was lower than in term infants [7]. Activated monocyte (by LPS) IL-6 production was lower in preterm infants than term infants [7]. Blood IL-6 produced by B-cells (stimulated with *Staphylococcus aureus*), T cells (stimulated with phytohemagglutinin) and monocytes (stimulated with LPS) were lower in preterm infants compared with term infants (these cells were isolated from their blood and tested in vitro) [19]. These results suggest a potential association between the reduction of IL-6 production by mononuclear cells in preterm infants and their higher susceptibility of preterm infants to bacterial infection. However, no study has compared the difference in cytokine production or immune-modulating activity between milk from preterm- and term-delivering mothers.

Preterm-delivering mothers often have difficulty expressing enough milk to meet their infant’s enteral nutrition needs and thus provide their infants supplemental donor milk (DM). DM is frozen and thawed at least two times and pasteurized at 65 °C for 30 min. Pasteurization reduces the concentration of immune proteins (including antibodies [20]) and freeze-thaw cycles and pasteurization can destroy immune cells. Therefore, DM may possess lower immunomodulatory properties compared with raw human milk, but no study has evaluated this hypothesis.

Cytokine gene expression by macrophages is often used to determine the immunomodulatory properties of a target molecule or mixture. In this study, the effect of milk from preterm- and term-delivering mothers as well as pooled, pasteurized DM on the cytokine gene expression (TNF-α, IL-6 and IL-12 (p40) and IL-10) by human macrophage-like cells derived from monocytic cell line THP-1 were compared. Gene expression for each gene of interest was measured via real-time reverse transcription polymerase chain reaction (RT-PCR) in the presence and absence of inflammatory stimuli. This research is a first step to understand the immunomodulatory properties of human milk immune proteins that may prevent infection and hyperinflammation (often associated with NEC) in newborn infants.

## 2. Results

Five filtered infranates from preterm-delivering mothers were pooled to create a pooled preterm milk (PM) and five others from term-delivering mothers were combined to make a pooled term milk (TM). Preterm-delivering mothers and term-delivering mothers had similar postpartum age for milk sampling (*p* = 0.39, Table 1). Maternal age tended to be lower in term-delivering mothers compared with that in preterm-delivering mothers. GA and infant birth weight were lower in preterm-delivering women than in term-delivering women (*p* < 0.001, Table 1). Pasteurized donor breast milk (DM) was a mixture of milks from donors with unknown demographics.

Immunomodulatory properties of PM, TM, DM and a control with no milk (Ctl) were compared using human macrophage-like cells derived from the monocytic cell line THP-1 at steady (uninflamed) and activated (inflammatory or inflamed) states. TNF-α, IL-6, IL-12 (p40), IL-10 and a housekeeping gene used for normalization (GAPDH) were measured via real-time RT-PCR.

TNF-α gene expression in steady state macrophages (non-inflammatory state, LPS 0 ng/mL) treated with DM was higher than in CTL-macrophages not treated with milk (*p* < 0.001) and treated with PM (*p* < 0.001) or TM (*p* = 0.042, Figure 1a). TNF-α expression did not differ between PM, TM and Ctl in the non-inflamed state. In the activated, inflammatory state (LPS 10 ng/mL), TNF-α gene expression increased for all sample types but this increase was significantly lower in macrophages treated with PM or TM compared with those treated with DM (*p* < 0.05) or untreated (Ctl, *p* < 0.001) (Figure 1b). In summary, DM increased pro-inflammatory TNF-α in the steady state and PM and TM reduced inflammation after inflammatory stimuli.

IL-6 gene expression in steady state macrophages treated with TM or DM was higher than that in Ctl-macrophages (*p* < 0.01) or macrophages treated with PM (*p* < 0.05, Figure 1c). IL-6 gene expression in activated macrophages treated with DM was higher than that in Ctl-macrophages (*p* < 0.01) or macrophages treated with PM or TM (Figure 1d). Overall, DM increased pro-inflammatory IL-6 in both uninflamed and inflamed macrophages while TM increased IL-6 expression in the uninflamed state.

IL-12 (p40) gene expression in steady state macrophages treated with TM was higher than that in Ctl-macrophages (*p* = 0.020) or macrophages treated with PM (*p* = 0.035, Figure 2a) but did not differ with DM. IL-12 (p40) gene expression also did not differ between Ctl, PM and DM. In the activated state, IL-12 (p40) gene expression increased in all groups, but its decrease was significantly less for PM (Figure 2b). In summary, TM increased pro-inflammatory IL-12 (p40) in the uninflamed state and PM prevented some degree of IL-12 (p40) induction in the inflamed state.

In the steady state, IL-10 gene expression in macrophage treated with TM or DM was higher than in Ctl-macrophages (*p* < 0.01) or macrophages treated with PM (*p* < 0.05, Figure 2c). In the activated state, IL-10 gene expression was increased across all groups and was significantly higher in macrophages treated with DM than in Ctl-macrophages (*p* = 0.048) or macrophages treated with TM (*p* = 0.032, Figure 2d). IL-10 expression between Ctl, PM and TM did not differ in activated macrophages. Therefore, TM and DM increased anti-inflammatory IL-10 in the uninflamed state and DM stimulated additional IL-10 production in the inflamed state. The overall results are illustrated in Figure 3.

## 3. Discussion

The activation of the immune responses in newborns is critical to appropriately protect against infectious microorganisms. Innate immune cells (macrophages, neutrophils and natural killer cells) recognize and become activated by molecules from pathogens, such as LPS from Gram-negative bacteria or teichoic acid from Gram-positive bacteria via pattern-recognition receptors. Activated macrophages can destroy pathogens by phagocytosis [1]. Moreover, innate immune cells can present antigens from a pathogen to naïve T cells, which will activate them into mature T cells. After recognition, mature T cells can activate B cells into plasma cells to produce antibodies specific to the pathogenic antigens [10]. During the first month of postnatal age, the clearance of pathogens is mainly performed by innate immune cells as the adaptive immune system of the newborn is still developing the adequate specificities, which might not be present or in low abundance [1]. Innate immune cells are present at lower numbers and are less chemotactic in newborns than in adults, which can also explain infants’ reduced ability to activate the adaptive immune system [21].

Immunomodulatory proteins can be transferred from mother’s milk to the neonate and reduce the risk of infection by enhancing their immune responses. However, whether the immunomodulatory effects differ between milk from preterm- and term-delivering mothers, and whether pasteurized DM has similar immunomodulatory actions on the innate immune cells across the neonatal gut remained unexplored. We compared the in vitro inflammatory cytokine response of macrophage-like cells derived from the monocytic cell line THP-1 to preterm milk, term milk and pasteurized milk with and without pre-exposure to LPS as an inflammatory stimulus. Although the in vitro THP-1 cell system cannot mimic the exposure of macrophages in the neonatal gastrointestinal tract during breastfeeding, our findings provide preliminary information about differences in pro-inflammatory and anti-inflammatory cytokine expression in macrophage-like cells induced by preterm milk, term milk and pasteurized milk. Expression and production of cytokines by immune cells can recruit other immune cells to a site and activate the immune response. The differences in cytokine expression induced by exposure to these different milks could indicate differences in how different milks can influence the immune response of infants. Monitoring how diet affects immune cell cytokine expression is particularly important for preterm infants who have immature immune systems and high risk for NEC development, which is associated with bacterial overgrowth and hyperinflammation [3]. This report is the first to examine how cytokine gene expression in human macrophage-like cells derived from the monocyte cell line THP-1 changes after treatment with PM, TM and DM compared to controls at steady and activated (inflammatory) states.

### 3.1. Unpasteurized Milks vs. Pasteurized Donor Milk

In the activated state, TNF-α gene expression in macrophages treated with PM or TM was lower than in macrophages not treated with milk (Ctl) or macrophages treated with DM. Likewise, in the activated state, IL-6 gene expression was reduced compared to the control after treatment with PM. These results indicate that PM and TM could help reduce the inflammatory response once it is activated by pathogens. Interestingly, immune proteins from raw human milk reduced macrophage TNF-α and IL-6 expression in the activated state, but not pasteurized human milk. Which immune components within these milks are responsible for this anti-inflammatory effect are unknown. These anti-inflammatory components could be proteins that are partially or fully denatured by pasteurization. Concentrations of antibodies and lactoferrin are well-known to be reduced during pasteurization of human milk [20,22,23], whereas concentrations of TGF-β and other cytokines are not affected and/or increased after heat-treatment [24,25]. Immune cells are likely not responsible to this anti-inflammatory effect due to their loss of viability during freezing and thawing of human milk. Bovine milk lactoferrin activated the secretion of TNF-α and IL-8 by macrophages from rat bone marrow (in vitro) [26], but this observation has not been evaluated in human macrophages. Phagocytosis of *Mycobacterium tuberculosis* by human macrophages increased in the presence of complement and tuberculosis-specific IgG antibody [27]. Therefore, we speculate that these differences of pro-inflammatory cytokine expression are mainly due to higher levels of antibodies and/or lactoferrin in raw human milk. These proteins (either from preterm-delivering mothers or term-delivering mothers) could potentially help to regulate the inflammatory response to pathogens in infants by reducing TNF-α and IL-6 production, especially for preterm infants at high risk to NEC [28]. Overexpression of inflammatory responses (TNF-α and IL-6) induced NEC in preterm rat model [29]. Development of NEC was associated with an exaggerated pro-inflammatory signaling during the activation of toll-like receptor 4 (TLR4) in the preterm gut. TLR4 can recognizes LPS of Gram-negative bacteria its overexpression is associated to induce NEC. Activation of TLR4 can increase enterocyte apoptosis, mucosal injury, intestinal ischemia and bacterial translocation, thus the development of NEC [30,31]. Animal models related to NEC also reported that higher pro-inflammatory cytokine levels increased the intestinal infiltration of macrophages in the premature intestine [30,31]. Higher expression of TNF-α in blood preterm infants was associated with higher risk of infections and NEC [32]. Higher TNF-α gene expression also increased the risk for sepsis, infection and NEC in a neonatal rat model [29,33]. Future studies are needed to evaluate the outcome of increasing pro-inflammatory cytokines IL-6 and TNF-α when infants are fed with pasteurized DM compared with raw human milk and whether these changes influence their overall immune responses and neonatal diseases.

IL-10 gene expression in macrophages treated with DM was higher compared with raw PM (steady state cells) and TM (activated cells). These observations in DM could be due to specific immune proteins that are more activated during heat-treatment. TGF-β2 bioactivity in both pasteurized DM and raw PM (at 1 week and 1 month of postpartum) was increased after heat-treatment at 80°C for 5 min [25]. These authors used this heat-treatment to activate latent TGF-β2. Latent TGF-β in milk is activated to the immunoreactive isoform by acidification (pH 3), proteases or heat-treatment, which denatures latency-associated peptide (LAP), a peptide that binds to TGF-β [34]. Pre-activation of human milk TGF-β2 using heat-treatment or acidification might be particularly critical for premature infants due to their higher gastric pH and lower protease activity in the stomach and intestine compared with term infants [35]. The capacity to activate TGF-β2 during digestion in preterm infants is unknown and future studies are needed to understand the functional activity of TGF-β2 in the neonatal gut.

### 3.2. Preterm vs. Term Milk

TM increased IL-6 and IL-12 (p40) gene expression by steady state macrophages compared to the control, but PM did not. In activated macrophages, PM reduced expression of IL-12 (p40), whereas TM did not. Whether this induction of IL-6 and IL-12 (p40) by TM in the uninflamed state and inhibition of IL-12 (p40) expression in the inflamed state occurs in vivo in breastfed infants remains unexplored. IL-6 can stimulate B cell proliferation and differentiation as well as T cell activation [9]. Higher IL-6 expression by T cells (Th2) promoted the induction and expression of IgA by plasma cells in a mouse intestine [36]. IL-6 may modulate IgA secretion as 60% decrease in total IgA plasma cells was observed in the intestinal lamina propria from IL-6 deficient mice [35]. However, early-onset neonatal infection at birth (24 h and 48 h of life) was associated with increasing IL-6 concentration in infant blood (independent of illness severity) [37]. The differences between TM and PM could be due to lower levels of anti-inflammatory or higher levels of pro-inflammatory proteins in TM compared with PM at 2–4 weeks of postpartum. IL-10, TGF-β and erythropoietin are present in human milk and possess anti-inflammatory effects by reducing excessive inflammatory response to stimuli in the infant’s gut [38]. Previous studies reported that IL-10 [39] and erythropoietin [40] concentrations did not differ between PM and TM during the first month of life. On the other hand, TGF-β concentration in milk from extremely preterm-delivery mothers (23–27 wk GA) were higher than that from preterm-(32–36 wk GA) or term-(38–42 wk GA) delivery mothers at 14 and 28 days of postnatal age [41]. TGF- β can control the excessive inflammatory responses by T-cells and stimulate IgA production. TGF-β1 inhibited the induction of scavenger receptor activity in macrophage-like cells derived from the monocytic cell line THP-1 [42]. The scavenger receptor can clear LPS during Gram-negative bacterial sepsis [42]. Therefore, higher level of TGF-β in PM may reduce macrophage phagocytosis activity, which could decrease the protective activity of innate immunity during neonatal infection. TGF-β2 was more efficient than TGF-β1 in suppressing inflammatory responses in the developing intestine and protecting against NEC (mice model) [43]. Orally ingested TGF-β2 protected baboons against NEC by suppressing mucosal inflammatory responses [44]. Higher TGF-β expression and bioactivity in animal and human intestinal tissues were associated with decreasing the risk of NEC [43]. TGF-β in PM could have an essential role to compensate for the immature immune system in premature infants to help prevent excessive inflammation in the gut, thus reducing NEC risk.

IL-10 gene expression in steady state macrophage-like cells treated TM was higher compared with PM. Milk IL-10 could promote the expansion of regulatory T cells in the mucosa [45] and enhances the production of SIgA and the epithelial integrity of the gut [46]. SIgA produced in the gut can improve the immune defense against infectious agents. IL-10 may also inhibit pro-inflammatory cytokine release (including IL-6 and TNF-α) and limit the activation of T_H_1 subset of helper T cells [46], which could reduce the systematic inflammation in the infant gut. Infants fed with PM may have a lower IL-10 expression by macrophages in the gut compared with those fed with TM, but no study has investigated this association.

A limitation of this study was the absence of the actual cytokine concentrations produced by macrophages (only gene expression). Whether the observed changes in cytokine expression have a direct effect on the amount of cytokine (actual protein production) that is release by the cells need to be determined in a future study. Moreover, the cytokine expression obtained with cancerous adult macrophages (THP-1) may differ from that obtained with neonatal macrophages when they are treated with human milk. Another limitation is that we only evaluated the cytokine production on macrophages, and other immune cells may respond differently to human milk.

This study selected the dose of 50 μg proteins/mL from one previous study that demonstrated changes in TNF-α gene expression when treated with a purified defensin peptide (LL-37) to THP-1 cells stimulated with 10 ng/mL of LPS [47]. We did preliminary experiments with 20, 50 and 100 μg proteins/mL for TNF-α gene expression with macrophage-like cells treated with PM. Lower standard variation between replicates was found when using 50 μg protein/mL of PM compared with the other protein concentrations (data not shown). The levels of defensins and other bioactive peptides (including cytokines, TLR and IgG) in transitional human milk vary between 0 and 50 μg/mL. Our future study will compare different peptide fractions from human milk at each step of purification to determine the effect of purified peptides on the cytokine gene expression by macrophage-like cells.

This present study demonstrates that milk from preterm- and term-delivering women, and pasteurized DM have different immunomodulatory effects on human macrophage-like cells. Whether these differences in cytokine expression lead to differing effects on overall immune responses in the neonatal gut should be investigated.

## 4. Materials and Methods

### 4.1. Participants and Sample Collection

This study was approved by the Institutional Review Boards of Legacy Health and Oregon State University (1402–2016, first approved on 05/03/17). Human milk samples were collected from 5 premature-delivering and 5 term-delivering mothers. These mothers delivered at 26 to 30 weeks of gestation and 38 to 40 weeks of gestation, respectively. Milk samples were collected from 7 to 30 days postpartum at the neonatal intensive care unit (NICU) (Table 1). Milk samples were collected by pumping on-site or at home with clean electric breast pumps into sterile vials (1.5 mL) and stored immediately at −20 °C. The breast was cleaned with water on a washcloth (no soap or alcohol) before pumping. Milk samples were transported to Oregon State University on dry ice and stored at −80 °C. Pasteurized donor breast milk (DM) was donated from Northwest Mother’s Milk Bank (Tigard, OR, USA).

### 4.2. General Sample Preparation

Human milk samples from each mother (PM and TM) and DM were thawed at 37 °C, centrifuged at 4226× *g* for 10 min at 4 °C, and the infranate was collected by pipette from below the upper fat layer. The five infranates (1 mL each) from preterm-delivering mothers were pooled (PM). The five infranates (1 mL each) from term-delivering mothers were combined to create a pooled term milk (TM). The PM, TM and DM supernatants were filtered through the syringe filters (GE Healthcare Whatman^TM^ Uniflo Syringe Filters, 0.22 µm, PES filter media, Thermo Fisher Scientific, Chino, CA, USA) under sterile conditions, aliquoted into sterile vials and stored at −80 °C until use in the immunomodulatory assays.

### 4.3. Immunomodulatory Assays

The immunomodulatory assays were performed following previous studies [48,49] with modifications. For the first experiment, immunomodulatory properties of PM, TM, DM and a control with no milk were compared using human macrophage-like cells derived from the monocytic cell line THP-1 at steady (uninflamed) and activated (inflammatory or inflamed) states. Three pro-inflammatory cytokines, tumor necrosis factor-alpha (TNF-α), (interleukin (IL)-6, IL-12 (p40), one anti-inflammatory cytokine IL-10 and a housekeeping gene used for normalization (GAPDH) were measured via real-time RT-PCR for all experiments.

### 4.4. Cell Isolation and Cell Lines

A new stock of monocytic cell line THP-1 (first passage) from the American Type Culture Collection (ATCC^®^ TIB-202^TM^, Manassas, VA, USA) was growth in suspension in Roswell Park Memorial Institute (RPMI) 1640 medium (Invitrogen^TM^, Grand Island, NY, USA) supplemented with 10% (*v*/*v*) heat-inactivated fetal bovine serum (FBS) (Invitrogen^TM^) in humidified incubators with 5% CO_2_ at 37 °C. The first passage of THP-1 cells (10 mL) were added in 10 mL of 90% FBS and 10% dimethyl sulfoxide (DMSO, EMD Millipore, Temecula, CA, USA), aliquoted into cryovial tubes (2 × 10^6^ cells per tube) and frozen progressively (−1 °C/min) to −80 °C overnight using a Mr. Frosty Freezing Container (Thermo Fisher Scientific). The day after, vials were transferred into a liquid nitrogen Dewar (−130 °C) for storage until experimentation.

### 4.5. Treatment with Inflammatory Stimuli and Human Milk

One THP-1 monocyte vial was thawed rapidly at 37 °C, added in a new tube containing 9 mL of RPMI, centrifuged at 1500× *g* for 10 min and the pellet was isolated from the supernatant. This washing step was repeated. THP-1 cells were seeded in tissue culture flasks at 3 × 10^6^ cells per T75 flask and differentiated into macrophages in RPMI media with 10% FBS containing 5 ng/mL of phorbol 12-myristate 13-acetate (PMA, Sigma-Aldrich, St-Louis, MO, USA) for 48 h. PMA was used to stimulate the activation and maturation of THP-1. After differentiation (cell morphology was changed from non-adherent (in suspension) to semi-adherent cells), media was aspirated, and THP-1 macrophages were conditioned for 24 h in RPMI with 10% FBS media. Protein concentration in PM, TM and DM was determined via BCA. Conditioned macrophages were treated with human milk solutions (PM, TM and DM diluted in RPMI media to a final concentration in the flask of at 50 μg proteins/mL in the well) or untreated (Ctl, no milk). After the one hour of milk incubation (without removing the milk solution), macrophages were treated with lipopolysaccharide (LPS) or left untreated and incubated for 3 h. Macrophages were stimulated with a low-dose of LPS (10 ng/mL final concentration in the well) from *Escherichia coli* O55:B5 (Sigma-Aldrich) to induce an inflammatory state (46). After 3 h, media in both untreated and treated macrophages were removed and washed one time with PBS. For the negative control, RPMI media was added on steady and activated Ctl-macrophages. All experiments using human THP-1 cells had at least 3 biological replicates.

### 4.6. Quantification of Cytokine Expression

Following incubation of the macrophages under various treatments, cells were washed one time with PBS and harvested. To harvest cells, 1 mL of PBS was added and wells were scraped with a cell scraper (Thermo Scientific) and then transferred to Eppendorf tubes. RNA was isolated from THP-1 cells by homogenization with the QIAshredder spin column and RNA purification with the RNeasy Mini Kit (Qiagen Sciences LLC, Louisville, KY, USA) (RNA lysate was treated in RNase-free DNase and eluted in RNase-free water (Thermo Fisher Scientific) as described the manufacturer’s instruction). RNA concentration and purity were determined using an ND-1000 UV-Vis spectrophotometer (Nanodrop Technologies, Thermo Fisher Scientific). One microgram total RNA was reverse transcribed into cDNA using SuperScript III First-Strand Synthesis SuperMix (Invitrogen, Pleasanton, CA, USA) for quantitative real-time PCR (qRT-PCR) using a thermal cycler at 50 °C for 30 min and heat inactivation at 85 °C for 5 min. Samples were plated on ice after the transcription. Real-time RT-PCR was performed using primers specific for human TNF-α, IL-6, IL-12 (p40), IL-10 and GAPDH (see Appendix A for the sequences of primers) made by Eurofins Genomics LLC (Louisville, KY, USA). Real-time RT-PCR reactions were performed with Fast SYBR Green Mastermix (Thermo Fisher Scientific) using ABI7900HT Fast Real-Time PCR System (Applied Biosystems, Foster City, CA, USA) using a standard cycling mode: 50 °C for 2 min (uracil-N-glycosylase gene (UNG) incubation), 95 °C for 2 min (polymerase activation) and then PCR (40 cycles) at 95 °C for 15 sec (denaturation) and 60 °C for 1 min (anneal/extend). Dissociation curve (melt curve stage) was performed as recommended by Life Technologies (95 °C for 15 sec, 60 °C for 15 sec and 95 °C for 15 sec). The dissociation curves were used to confirm a single peak for each specific amplification (absence of nonspecific amplification or primer-dimer formation). Gene copies were determined using the standard curve method. A standard curve was generated from serial dilutions of purified plasmid DNA that encoded for each gene of interest. Data represent averaged fold-change of milk-treated THP-1 cells compared to untreated THP-1 cells (no milk) after the copy number of the gene of interest was normalized to the copy number of GAPDH housekeeping gene.

### 4.7. Statistical Analyses

One-way ANOVA followed by Tukey’s multiple comparisons test was used to compare gene expression (fold-change) for each cytokine between Ctl, PM, TM and DM using GraphPad Prism version 8.0 (GraphPad Software Inc., San Diego, CA, USA). Unpaired *t* tests were used to compare the characteristics between preterm-delivering women and term-delivering women. Differences were designated significant at *p* < 0.05. This study used a mixture of human milk samples (*n* = 5 per group) from five preterm-delivery women and five term-delivery women, which allows to reduce some variability between individuals and detect significant differences related to premature birth for cytokine gene expression between preterm-group and term-group. The number of individual selected proved to be adequately powered based on the result.

## 5. Conclusions

This study demonstrates that milk from preterm- and term-delivering women, and pasteurized DM have different immunomodulatory effects on human macrophages-like cells derived from the monocytic cell line THP-1. DM increased pro-inflammatory TNF-α in the non-inflamed state whereas PM and TM reduced inflammation after inflammatory stimuli. DM increased pro-inflammatory IL-6 in both steady and inflamed macrophages while TM increased IL-6 expression in the uninflamed state. TM increased pro-inflammatory IL-12 (p40) in the steady state and PM prevented some degree of IL-12 (p40) induction in the inflamed state. TM and DM increased anti-inflammatory IL-10 in the uninflamed state and DM stimulated additional IL-10 production in the inflamed state. Whether these differences in cytokine expression lead to differences in the innate and adaptive immune response against microbial infection in the neonatal gut need to be investigated. Higher pro-inflammatory cytokine expression in pasteurized DM compared with PM or TM need to be evaluated in a clinical study to determine whether this increase of inflammatory cytokines may affect the systematic inflammation in the neonatal gut.

## Figures and Tables

**Figure 1 molecules-25-02376-f001:**
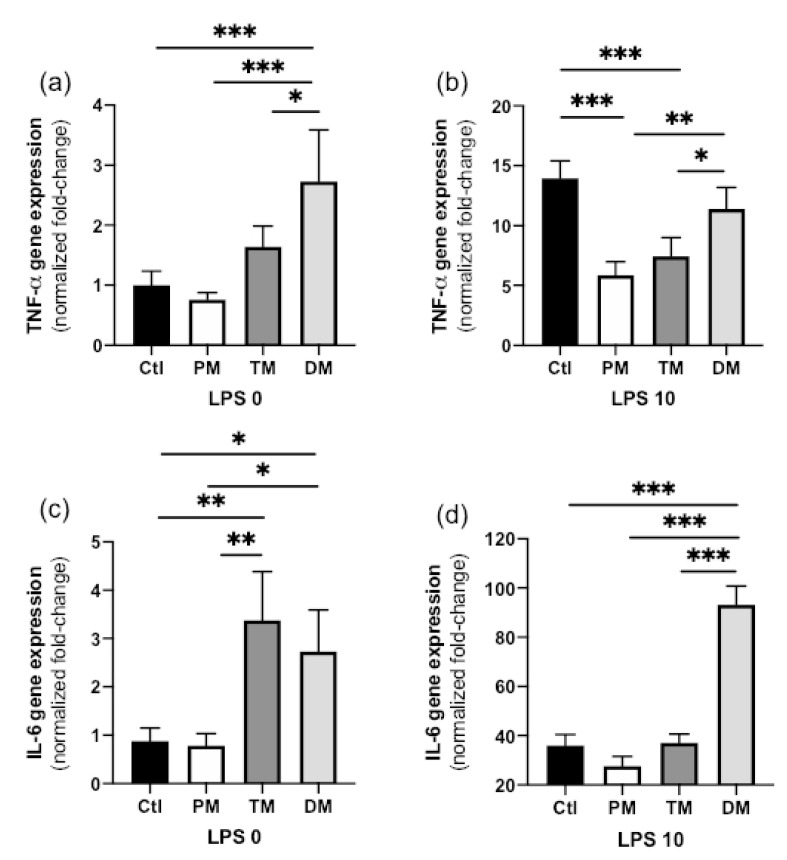
(**a**,**b**) Tumor necrosis factor (TNF)-α and (**c**,**d**) interleukin-6 (IL-6) gene expression by human macrophage-like cells derived from the monocytic cell line THP treated with pooled milk from preterm-delivering mothers (PM), term-delivering mothers (TM), pasteurized donor milk (DM) or in absence of milk (Ctl). Macrophage-like cells were incubated with the treatments at 50 μg total proteins/mL or no treatment (media) for 1 h and then incubated (**a**,**c**) at steady state (LPS 0 ng/mL) for 3 h or (**b**,**d**) activated with low-dose lipopolysaccharides (LPS) (10 ng/mL, low inflammatory state) for 3 h. Values are mean ± SD of three independent experiments and three replicates in each experiment. Fold-changes for each gene were normalized to GAPDH and are relative to the gene expression in untreated cells (without LPS or milk). Asterisks show statistically significant differences between groups (*** *p* < 0.001; ** *p* < 0.01; * *p* < 0.05) using one-way ANOVA followed by Tukey’s multiple comparisons tests.

**Figure 2 molecules-25-02376-f002:**
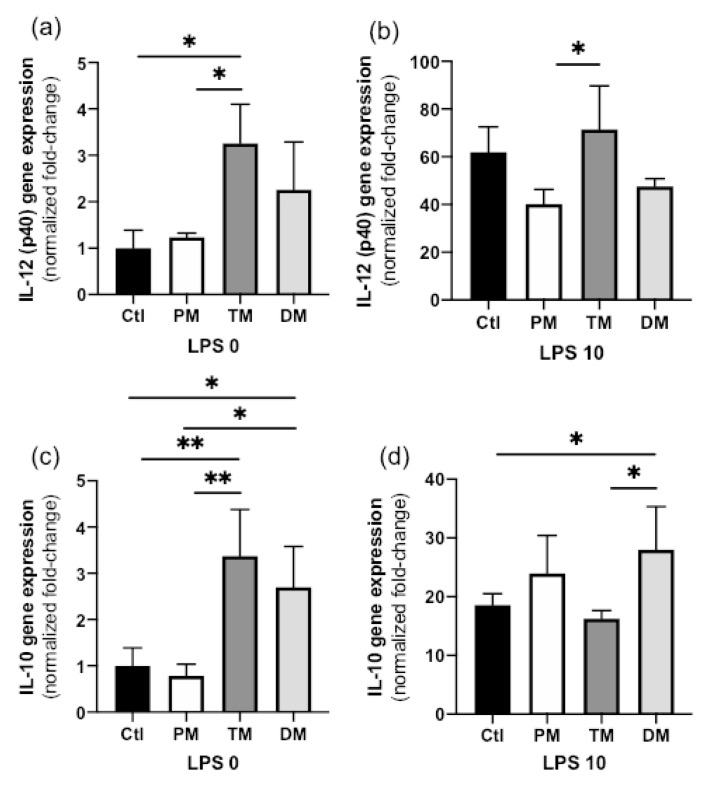
(**a**,**b**) Interleukin-12 (IL-12) (p40) and (**c**,**d**) interleukin-10 (IL-10) gene expression by human macrophage-like cells derived from the monocytic cell line THP-1 treated with pooled milk from preterm-delivering mothers (PM), term-delivering mothers (TM), pasteurized donor milk (DM) or in absence of milk (Ctl). Macrophages were incubated with the treatments at 50 μg total proteins/mL or no treatment (media) for 1 h and then incubated (**a**,**c**) at steady state (LPS 0 ng/mL) for 3 h or (**b**,**d**) activated with low-dose lipopolysaccharides (LPS) (10 ng/mL, low inflammatory state) for 3 h. Values are mean ± SD of three independent experiments and three replicates in each experiment. Fold-changes for each gene were normalized to GAPDH and are relative to the gene expression in untreated cells (without LPS or milk). Asterisks show statistically significant differences between groups (** *p* < 0.01; * *p* < 0.05) using one-way ANOVA followed by Tukey’s multiple comparisons tests.

**Figure 3 molecules-25-02376-f003:**
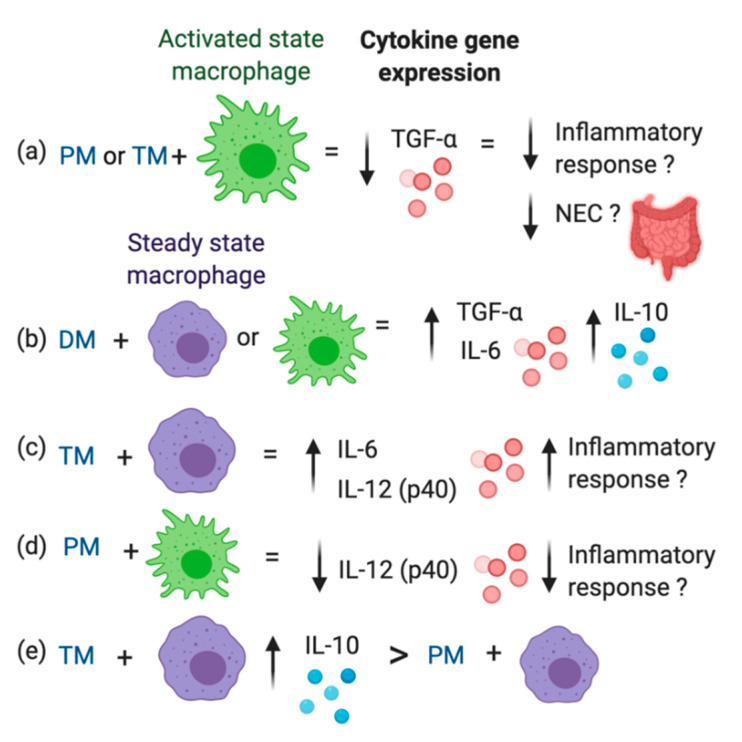
Overview of cytokine gene expression by macrophage-like cells derived from the monocytic cell line THP-1 treated with pooled milk from preterm-delivering mothers (PM), term-delivering mothers (TM), or pasteurized donor milk (DM). (**a**) In the activated state, TNF-α gene expression in macrophages treated with PM or TM was lower than in macrophages not treated with milk (control) or macrophages treated with DM. (**b**) IL-10 gene expression in macrophage-like cells treated with DM was higher compared with raw PM (steady state cells) and TM (activated cells). (**c**) TM increased IL-6 and IL-12 (p40) gene expression by steady state macrophages compared to the control, but PM did not. (**d**) In activated macrophages, PM reduced expression of IL-12 (p40), whereas TM did not. (**e**) IL-10 gene expression in steady state macrophage-like cells treated TM was higher compared with PM.

**Table 1 molecules-25-02376-t001:** Characteristics of preterm- and term-delivering mothers for milk collection ^1^.

Characteristics	Preterm	Term	*p*-Value
GA, wks	24.8 ± 0.2 (24–26)	38.6 ± 0.9 (37–39)	<0.001
Postpartum age, days	23 ± 6 (17–28)	28 ± 10 (14–42)	0.39
Infant birth weight (kg)	0.70 ± 0.06 (0.62–0.76)	3.6 ± 0.3 (3.0–3.8)	<0.001
Infant gender, female:male (*n*)	3:2	1:4	-
Mother age	32 ± 1 (31–34)	23 ± 11 (17–42)	0.084

^1^ Data are mean ± SD (min–max) for milk samples from preterm-delivery mothers (*n* = 5) and term-delivery mothers (*n* = 5). *p*-value was determined using unpaired *t* test.

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
