# Peer review of "Cytokine Expression by Human Macrophage-Like Cells Derived from the Monocytic Cell Line THP-1 Differs between Treatment with Milk from Preterm- and Term-Delivering Mothers and Pasteurized Donor Milk"

_molecules, 2020, doi:10.3390/molecules25102376_

Round 1

Reviewer 1 Report

In the manuscript „Human macrophage cytokine expression differs between treatment with milk from preterm- and term-delivering mothers and pasteurized donor milk“ by Veronique Demers-Mathieu and colleagues, the authors investigate how milk from preterm-(PM) mothers, term-delivering (TM) mothers and pasteurized donor milk (DM) induces changes in expression levels of different cytokines in human macrophages. They report different effects on expression of TNFα, IL-6, IL-10 and IL-12.

The manuscript is easy to follow. The data are clearly presented and discussed. However, the whole study consists of a few RT-PCR measurements and gives the impression of a rather preliminary data set and not a full scientific paper. Furthermore, the following points reduce my enthusiasm for the study:

  • The authors write that they have used human macrophages (which gives the reader the impression that primary cells have been used), but instead used macrophage-like cells derived from the monocytic cell line THP-1. This has to be corrected in the manuscript, including the title.
  • The cell system used (macrophage-like cells with and without stimulus by LPS) appears rather artificial. It is unclear to this reviewer how this relates to something that is actually happening in the infant.
  • Line 88: „Preterm-delivering mothers and term-delivering mothers had similar characteristics“ Appropriate statistical analyses should be used to determine whether the two groups are similar or different regarding the characteristics presented in table 1.
  • It is unclear how the number of participants (n=5 per group) that are needed to get significant results was determined. With only five donors per group, the study appears to be underpowered.
  • Figure 2: IL-12 is a heterodimeric cytokine that consists of two individual proteins encoded by two separate genes. It is therefore not possible to investigate IL-12 gene expression by RT-PCR. Detection of IL-12 has to be done by a different method, e.g. by Western blot or ELISA from the cell supernatant.  
  • The authors only report relative changes in cytokine expression. It is unclear how this translates into actual protein production, and whether the observed changes have an direct effect on the amount of cytokine that is released by the cells.

Author Response

In the manuscript „Human macrophage cytokine expression differs between treatment with milk from preterm- and term-delivering mothers and pasteurized donor milk“ by Veronique Demers-Mathieu and colleagues, the authors investigate how milk from preterm-(PM) mothers, term-delivering (TM) mothers and pasteurized donor milk (DM) induces changes in expression levels of different cytokines in human macrophages. They report different effects on expression of TNFα, IL-6, IL-10 and IL-12.

The manuscript is easy to follow. The data are clearly presented and discussed. However, the whole study consists of a few RT-PCR measurements and gives the impression of a rather preliminary data set and not a full scientific paper. Furthermore, the following points reduce my enthusiasm for the study:

  • The authors write that they have used human macrophages (which gives the reader the impression that primary cells have been used), but instead used macrophage-like cells derived from the monocytic cell line THP-1. This has to be corrected in the manuscript, including the title.

>> Thank you, we changed “human macrophage” for “human macrophage-like cells derived from the monocytic cell line THP-1”. 

  • The cell system used (macrophage-like cells with and without stimulus by LPS) appears rather artificial. It is unclear to this reviewer how this relates to something that is actually happening in the infant.

>> We added a paragraph to improve the clarity of the cell system used (macrophage-like cells with and without stimulus by LPS) to relate to the infant. We compared the in vitro inflammatory cytokine response of macrophage-like cells derived from the monocytic cell line THP-1 to preterm milk, term milk and pasteurized milk with and without pre-exposure to LPS as an inflammatory stimulus. Although the in vitro THP-1 cell system cannot mimic the exposure of macrophages in the neonatal gastrointestinal tract during breastfeeding, our findings provide preliminary information about differences in pro-inflammatory and anti-inflammatory cytokine expression in macrophage-like cells induced by preterm milk, term milk and pasteurized milk. Expression and production of cytokines by immune cells can recruit other immune cells to a site and activate the immune response. The differences in cytokine expression induced by exposure to these different milks could indicate differences in how different milks can influence the immune response of infants. Monitoring how diet affects immune cell cytokine expression is particularly important for preterm infants who have immature immune systems and high risk for NEC development, which is associated with bacterial overgrowth and hyperinflammation (line 187-199)

  • Line 88: „Preterm-delivering mothers and term-delivering mothers had similar characteristics“ Appropriate statistical analyses should be used to determine whether the two groups are similar or different regarding the characteristics presented in table 1.

>> We added additional statistical analyses to determine whether the two groups are similar or different regarding the characteristics in table 1. (line 97-101, 390-391)

  • It is unclear how the number of participants (n=5 per group) that are needed to get significant results was determined. With only five donors per group, the study appears to be underpowered.

>> This study used a mixture of human milk samples (n = 5 per group) from 5 preterm-delivery women and 5 term-delivery women, which allows to reduce some variability between individuals and detect significant difference related to prematurity (gestational age) for cytokine gene expression between preterm-group and term-group. The number of individual selected proved to be adequately powered based on the result. We added this information in the statistical analyses. (line 392-396)

  • Figure 2: IL-12 is a heterodimeric cytokine that consists of two individual proteins encoded by two separate genes. It is therefore not possible to investigate IL-12 gene expression by RT-PCR. Detection of IL-12 has to be done by a different method, e.g. by Western blot or ELISA from the cell supernatant.  

>>Thank you for this great comment. We forgot to indicate human IL-12 (p40) (Genbank accession#: AF180563) in the manuscript. We changed IL-12 for IL-12 (p40).

We also find several manuscripts that performed the gene expression of IL-12 (p40) by RT-PCR. Per example:

Chen, R.F., Wang, L., Cheng, J.T. and Yang, K.D., 2012. Induction of IFNα or IL-12 depends on differentiation of THP-1 cells in dengue infections without and with antibody enhancement. BMC infectious diseases12(1), p.340.(https://bmcinfectdis.biomedcentral.com/articles/10.1186/1471-2334-12-340).

  • The authors only report relative changes in cytokine expression. It is unclear how this translates into actual protein production, and whether the observed changes have an direct effect on the amount of cytokine that is released by the cells.

>> Thank you. We added at the end of the discussion this limitation (line 287-289).

Reviewer 2 Report

In their manuscript, Demers-Mathieu et al. test the effect of milk from preterm- and term-delivered mothers on the secretion of cytokines by macrophages and compare it to the effect by pasteurized donor milk. They demonstrate differential immune modulatory effects on macrophage activities. Although the outcome of the study is still rather heterogeneous, it could be considered as initiation of more intensive work on effectors of immune modulators in such milk preparations. Therefore, the work is publishable provided some amendments to the manuscript are added.

  1. English need improvement in grammar and style.
  2. The authors claim that macrophages were tested, however they use the differentiated cell line THP-1. This is acceptable but should be mentioned in the summary and text of results.
  3. The macrophages were not inactivated, rather they were not activated or in steady state. This is confusing and should be corrected.
  4. The cytokines mentioned in the introduction have very pleiotropic effects. However, the author restrict the description to a single property each. This maybe the most relevant for the present work. Nevertheless, this needs to be clarified by appropriate statements.
  5. In the results, the assay used for determination the effects of the different milk preparations on macrophages needs to be described shortly. Otherwise, it is not clear how the data were established. M&M comes later and the details are ok but for understanding an earlier short description would help the reader.
  6. The authors use “CTL” as abbreviation for control. However, in mmuno,logy CTL stands for cytotoxic T lymphocytes. “Ctr” would be more appropriate.
  7. The statistics is very unclear. Using “a” and “b” is very unusual. In the legend once both letters indicate p<0.05. It is also not clear what is compared with what. Commonly * or ** are used and the bars compared are connected by a line.
  8. Line 151 if the t and B cells would be naïve the immune system of the newborn could aüppriprately respond to a challenge, especially since antibodies from the mother would be present. The problem of the newborn is that the adaptive immune system is still developing the correct specificities might not be present or in a too low frequency. Please correct.
  9. The authors completely ignore the transfer of antibodies from the mother to the fetus via the placenta. Please extend.
  10. Line 243: I assume the authors meant TNF-a instead of TGF-a.

Author Response

In their manuscript, Demers-Mathieu et al. test the effect of milk from preterm- and term-delivered mothers on the secretion of cytokines by macrophages and compare it to the effect by pasteurized donor milk. They demonstrate differential immune modulatory effects on macrophage activities. Although the outcome of the study is still rather heterogeneous, it could be considered as initiation of more intensive work on effectors of immune modulators in such milk preparations. Therefore, the work is publishable provided some amendments to the manuscript are added.

  1. English need improvement in grammar and style.

>> We improved the grammar and style of the manuscript.

  1. The authors claim that macrophages were tested, however they use the differentiated cell line THP-1. This is acceptable but should be mentioned in the summary and text of results.

>> Thank you, we changed “human macrophage” for “human macrophage-like cells derived from the monocytic cell line THP-1”. 

  1. The macrophages were not inactivated, rather they were not activated or in steady state. This is confusing and should be corrected.

>> Thank you for this great comment, we changed “not inactivated” to in steady state in the manuscript.

  1. The cytokines mentioned in the introduction have very pleiotropic effects. However, the author restrict the description to a single property each. This maybe the most relevant for the present work. Nevertheless, this needs to be clarified by appropriate statements.

>> Thank you. We added “These cytokines have pleiotropic effects by modulating the activity of multiple cell types” in the introduction (line 54-55).

  1. In the results, the assay used for determination the effects of the different milk preparations on macrophages needs to be described shortly. Otherwise, it is not clear how the data were established. M&M comes later and the details are ok but for understanding an earlier short description would help the reader.

>>Thank you. We added a short description in the beginning of the result section to help the ready to understand the study. (line 95-97, line 105-108)

  1. The authors use “CTL” as abbreviation for control. However, in immunology CTL stands for cytotoxic T lymphocytes. “Ctr” would be more appropriate.

>> We changed “CTL” for “Ctl” in the paper.

  1. The statistics is very unclear. Using “a” and “b” is very unusual. In the legend once both letters indicate p<0.05. It is also not clear what is compared with what. Commonly * or ** are used and the bars compared are connected by a line.

>> Thank you for this good comment. We changed the figures to use *, **, or *** to compare the groups.

  1. Line 151 if the t and B cells would be naïve the immune system of the newborn could appropriately respond to a challenge, especially since antibodies from the mother would be present. The problem of the newborn is that the adaptive immune system is still developing the correct specificities might not be present or in a too low frequency. Please correct.

>>Thank you, we corrected this sentence to clarify the explanation (line 179-180).

  1. The authors completely ignore the transfer of antibodies from the mother to the fetus via the placenta. Please extend.

>>We added one sentence in the introduction to extend the effect of prematurity on the transfer of IgG from mother to the fetus via placenta (line 39-41).

  1. Line 243: I assume the authors meant TNF-a instead of TGF-a.

>>Thank you, we corrected this mistake (line 282).

Reviewer 3 Report

The manuscript by Demers-Mathieu et al. entitled “Human macrophage cytokine expression differs between treatment with milk from preterm- and term-delivering  mothers and pasteurized donor milk” describes the immunomodulatory effects of different human milk types on THP-1 cells. The experiments are well performed, the manuscript is well analyzed, interpreted and written. We only have minor comments as detailed below that need to be addressed by the authors. 

  1. How did the authors establish the dose of 50 ug proteins/ml? Was a titration performed to determine the optimal response? If yes, it would be important to include this data in the manuscript. Also, how relevant is this dose to overall quantity of milk ingested by the infant?  
  2. Although the authors write that they use THP-1 cells, this information is not completely evident in the manuscript. In order to avoid confusion, especially to untrained eyes, it needs to be more clearly stated that THP-1 cell line was used in the abstract, the introduction and conclusion. For example:
    1. In the abstract on line 16 please modify from: “... on cytokine expression in human macrophages.” to “... on cytokine expression levels in human THP-1 monocyte cell line.”
    2. In the introduction on line 81 please modify from: “... DM on human macrophage cytokine...” to “... DM on human differentiated THP-1 macrophage cytokine...”.
  3. The discussion is highly hypothetical and could be more concise:
    1. the authors describe several other studies on immune effects of different milk-containing substances that potentially might be responsible for the effects they are reporting, however they did not assess the levels of these molecules in their milk samples (TGF-b1/2/3, lactoferrin, IL-10, antibodies, erythropoietin, ...); without this information these sections are too speculative and should be shortened.
    2. The authors allude in some instances on effects on adaptive immune cells (like T cells, on lines 214-216, 240-241), however how pertinent are these comments in context of their results with macrophages/ innate immune cells? Is there any data available on NK cells, PMNs, DCs, ...?
    3. Animal studies are a little bit mixed with human observations, these need to be more clearly separated. To what extent are the animal models of NEC dependent on monocytes/macrophages or other innate immune cells?
  4. Errors:
    1. Line 158-159 a verb is missing.
    2. Line 188 verb tense is inappropriate.
    3. Line 189 please remove “pathogen” as TLR4 recognizes LPS derived from bacteria that can be pathogenic but also not necessarily so.
    4. Line 192 reference 9 seems to be out of place.
    5. Line 203, please insert “that” after “... a peptide binds to TGF-b.”
    6. Line 231-232, the sentence about 3 TGF-b isoforms seems out of place and incomplete. TGF-b2 was already mentioned above on lines 204-208. Is there any difference between these 3 isoforms?
    7. In M&M line 303, the 1 hr of milk incubation is missing.       

Author Response

The manuscript by Demers-Mathieu et al. entitled “Human macrophage cytokine expression differs between treatment with milk from preterm- and term-delivering mothers and pasteurized donor milk” describes the immunomodulatory effects of different human milk types on THP-1 cells. The experiments are well performed, the manuscript is well analyzed, interpreted and written. We only have minor comments as detailed below that need to be addressed by the authors. 

  1. How did the authors establish the dose of 50 ug proteins/ml? Was a titration performed to determine the optimal response? If yes, it would be important to include this data in the manuscript. Also, how relevant is this dose to overall quantity of milk ingested by the infant?

>>Thank you for this great comment. We selected 50 ug/mL protein from a previous paper that demonstrated changes in gene expression for TNF-alpha when added peptide LL-37 to THP-1 cells stimulated with LPS10. (Mookherjee N, et al. Modulation of the TLR-mediated inflammatory response by the endogenous human host defense peptide LL-37. J Immunol. 2006 15;176(4):2455-64). We did a preliminary experiment with 20, 50, 100 ug/mL for TNF-alpha with PM but only two repetitions using duplicate. Lower standard variation between replicates was found when using 50 ug protein/mL compared with the other protein concentrations. The relevance of this dose to the overall quantity of milk ingested by the infant needs to be investigated. We added this information to the limitation section (line 293-301).

  1. Although the authors write that they use THP-1 cells, this information is not completely evident in the manuscript. In order to avoid confusion, especially to untrained eyes, it needs to be more clearly stated that THP-1 cell line was used in the abstract, the introduction and conclusion. For example:
    1. In the abstract on line 16 please modify from: “... on cytokine expression in human macrophages.” to “... on cytokine expression levels in human THP-1 monocyte cell line.”
    2. In the introduction on line 81 please modify from: “... DM on human macrophage cytokine...” to “... DM on human differentiated THP-1 macrophage cytokine...”.

>> Thank you, we changed “human macrophage” for “human macrophage-like cells derived from the monocytic cell line THP-1” everywhere in the manuscript.

  1. The discussion is highly hypothetical and could be more concise:
    1. the authors describe several other studies on immune effects of different milk-containing substances that potentially might be responsible for the effects they are reporting, however they did not assess the levels of these molecules in their milk samples (TGF-b1/2/3, lactoferrin, IL-10, antibodies, erythropoietin, ...); without this information these sections are too speculative and should be shortened.

>> The differences of level for TGF, lactoferrin, IL-10, antibodies, …) between preterm, term and pasteurized donor milk have been previously well-described in several previous studies. We cited those studies to help understanding our results.

    1. The authors allude in some instances on effects on adaptive immune cells (like T cells, on lines 214-216, 240-241), however how pertinent are these comments in context of their results with macrophages/ innate immune cells? Is there any data available on NK cells, PMNs, DCs, ...?

>>We indicate some possible effects of IL-6 on T cells and B cells as we observed changes in IL-6 gene expression by macrophages between preterm, term and pasteurized milk. Most previous studies observed that IL-6 affected T cells and B cells (line 253-255).

    1. Animal studies are a little bit mixed with human observations, these need to be more clearly separated. To what extent are the animal models of NEC dependent on monocytes/macrophages or other innate immune cells?

>>We added that animal models related to NEC have observed that high intestinal infiltration of macrophages in the premature intestine could play a role in NEC development (line 228-230).

  1. Errors:
    1. Line 158-159 a verb is missing.

>> We added “is” in the sentence (line 199).

    1. Line 188 verb tense is inappropriate.

>>We corrected the verb tense (“was” associated) (line 224)

    1. Line 189 please remove “pathogen” as TLR4 recognizes LPS derived from bacteria that can be pathogenic but also not necessarily so.

>>We removed “pathogen” (line 226).

    1. Line 192 reference 9 seems to be out of place.

>>Thank you. We removed this reference (mistake) from the sentence (line 231).

    1. Line 203, please insert “that” after “... a peptide binds to TGF-b.”

>>We added “that” before “binds to TGF-b.” (line 242)

    1. Line 231-232, the sentence about 3 TGF-b isoforms seems out of place and incomplete. TGF-b2 was already mentioned above on lines 204-208. Is there any difference between these 3 isoforms?

>> We changed these sentences to improve the overall explanation. We indicated that a previous study demonstrated that TGF-b2 was more efficient than TGF-b1 in suppressing inflammatory responses in the developing intestine and protecting against NEC (mice model) (line 269-275).

    1. In M&M line 303, the 1 hr of milk incubation is missing. 

>>We added “the 1 h of “ before milk incubation. (line 254)

Reviewer 4 Report

Excellent and important research question.  It would be important to look at the cytokines themselves, and the temporal nature of secretion.  Stimulation over time with LPS, different concentrations of LPS.  This is a good start.  

Author Response

Excellent and important research question.  It would be important to look at the cytokines themselves, and the temporal nature of secretion.  Stimulation over time with LPS, different concentrations of LPS.  This is a good start.  

>> Thank you for your comment. We agree that a future study is needed to determine the cytokine concentrations as well as more experiments with stimulation over time with LPS at different concentrations.

Round 2

Reviewer 1 Report

The authors have responded well to my criticism and added additional and missing information. While I still think that this is rather a preliminary dataset and not a full scientific paper, I have no further objections.